# Influence of Graphite Powder on the Mechanical and Acoustic Emission Characteristics of Concrete

**Wei He [1,\*], Wenru Hao [1], Xia Meng [2], Pengchong Zhang [1], Xu Sun [1] and Yinlan Shen [3,4]**

[1] School of Civil and Transportation Engineering, Beijing University of Civil Engineering and Architecture, Beijing 100044, China; haowenru1@163.com (W.H.); zhangpengchong@bucea.edu.cn (P.Z.); sunxu@bucea.edu.cn (X.S.)

[2] Architectural Design and Research Institute of Tsinghua University Co., Ltd., Beijing 100084, China; mengxia@thad.com.cn

[3] Faculty of Architecture, Civil and Transportation Engineering, Beijing University of Technology, Beijing 100124, China; shenyinlan@bjut.edu.cn

[4] College of Materials Science and Technology, Beijing Forestry University, Beijing 100083, China

\* Correspondence: hewei@bucea.edu.cn

**Abstract:** In this paper, uniaxial compressive strength (UCS) test and three-point bending (TPB) test, together with an acoustic emission (AE) system, were performed to investigate the mechanical properties and AE characteristic changes of concrete with different graphite powder (GP) content. The results show that: (1) Poor adhesion and low interlocking of graphite with cement stone increase the initial defects of concrete, reducing its elastic modulus and the cyclo-hoop effect, and thus weakening the compressive strength. (2) For concrete with a low graphite content, the second sharp rise in ringing counts or energy released during the compressive process can be regarded as a failure alarm. However, as GP content increases, the second sharp rise fades away, while the first sharp rise becomes more visible. At high GP content, the first sharp rise is better for predicting failure. (3) The initial defects caused by GP significantly lower the initial fracture toughness, but its bridging effect greatly increases the critical crack mouth opening displacement and thus significantly enhances the unstable fracture toughness of concrete, by up to 9.9% at 9% GP content. (4) In contrast to compressive process, the sharp increase in AE signals preceding failure during the fracture process cannot be used to predict failure because it occurs too close to the ultimate load. However, as GP can significantly increase the AE signals and damage value in the stable period, such failure precursor information can provide a safety warning for damage development.

**Keywords:** graphite concrete; mechanical properties; acoustic emission; failure prediction

## 1. Introduction

Electrically conductive concrete (ECC) is a specialized category of concrete made by adding electrically conductive components into conventional concrete. After adding conductive components, the electrical conductivity of concrete can be increased to the necessary level for various applications, such as melting snow [1], cathodic protection [2], grounding electrodes [3], and electromagnetic interference shielding [4]. Moreover, owing to the mechano-electric effect of ECC, it can be used in traffic monitoring [5] and health monitoring for large structures [6]. When used in monitoring, the volume of ECC must be large enough to provide adequate coverage; thus, it should be regarded as an essential component of the bearing structure. Sometimes, a local failure may not lead to a significant change in electrical characteristics of ECC, but will cause the collapse of the entire structure. Thus, additional means, such as acoustic emission (AE), should be used to monitor potential local damage.

Acoustic emissions are the elastic waves generated by microstructural damage within a material. AE-based techniques utilize AE sensors to detect the released energy of local

deformation and micro-crack development, and reflect the degree of damage to the material [7]. By analyzing AE signals, the failure process of concrete can be quantified, thus effectively describing the damage mechanism [8]. In recent years, based on time-domain features and spectral characteristics [9], after studying the relationship between stress release during failure stage and precursor information by AE parameters, Xiao et al. [10] proposed a prediction method for the dynamic failure of rock. Lai et al. [11] found that a low-frequency energy percentage above a certain threshold can be regarded as the damage critical point, and that it can be used as the failure precursor criterion for stability monitoring and early warning of concretes.

Generally, conductive fillers used in ECC include graphite powder, coke breeze, steel shavings, and steel or carbon fibers. Graphite powder is one of the most commonly used conductive fillers for ECC due to its low cost, excellent conductivity, and chemical stability. However, due to its high specific surface area, flake shape, and poor adhesion with cement stone, the mechanical properties and failure pattern of concrete are significantly changed. Topolář et al. [12] compared the effect of carbon black and graphite powder on the acoustic emission parameters during three-point bending tests, and it was found that an acoustic emission method was able to detect failure at a very early stage, long before a structure completely fails, and that the acoustic emission characteristics of concrete vary with graphite powder dosage. However, there is a lack of extended research into the early warning time and the trends in damage development at various filler dosages.

When used as conductive filler, graphite powder can be either used alone or in combination with other conductive fillers. Because of the percolation threshold effect, when graphite powder is used alone, a larger dosage is required to confer good conductivity to the concrete. When graphite powder is combined with carbon fibers to form conductive frameworks, the amount of graphite required is relatively low. Using these two dosages of graphite powder as research objects, this paper aims to reveal the influence of graphite powder on the mechanical and AE characteristics of concrete from the compression test and three-point bending test, along with the AE test, to guide improvements in the ability of AE detection systems to predict the failure of graphite concrete. The research results will support the use of graphite powder in conductive concrete in important structures such as dams and nuclear facilities.

## 2. Experimental Details

### 2.1. Materials

The cement used for testing was ordinary Portland cement P.O 42.5 produced by Beijing JinYu Group Co., Ltd. The chemical components and physical properties are shown in Tables 1 and 2. Natural flake graphite powder (grade 5000 mesh) purchased from Yanhai Carbon Materials Co., Ltd. (Qingdao, China) was adopted, and its micromorphology is shown in Figure 1.

**Table 1.** Cement chemical components/%.

| CaO | SiO$_2$ | Al$_2$O$_3$ | MgO | Fe$_2$O$_3$ | SO$_3$ | Na$_2$O | K$_2$O | Loss |
|---|---|---|---|---|---|---|---|---|
| 58.51 | 21.05 | 7.90 | 3.78 | 2.97 | 2.73 | 0.44 | 0.85 | 1.77 |

**Table 2.** Main physical properties of cement.

| Density (g/cm$^3$) | Specific Surface Area (m$^2$/kg) | Setting Time (min) | | Flexural Strength (MPa) | | Compressive Strength (MPa) | |
|---|---|---|---|---|---|---|---|
| | | Initial | Final | 3 d | 28 d | 3 d | 28 d |
| 2.94 | 315 | 160 | 245 | 5.2 | 9.0 | 25.4 | 50.2 |

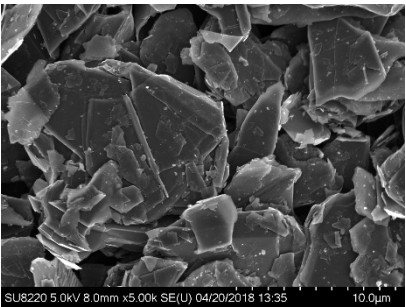

**Figure 1.** SEM microtopography of graphite powder.

Two sizes of specimen were prepared, one was a cubic sample (100 × 100 × 100 mm³), the other was a concrete beam (400 × 100 × 100 mm³) with a precast notch of 2 mm width and 40 mm height in the middle of the sample (Figure 2). The mix proportions of the concrete are shown in Table 3.

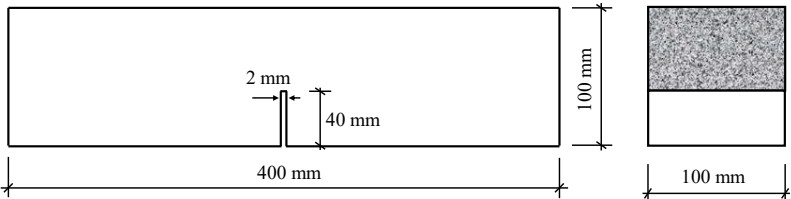

**Figure 2.** Schematic diagram of the concrete beam with precast notch.

**Table 3.** Mix proportion of the concrete/kg/m³.

| No. | Specimen Shape | Cement | Coarse Aggregate | Fine Aggregate | | Water |
| --- | --- | --- | --- | --- | --- | --- |
| | | | | River Sand | Graphite Powder | |
| C1 | | 433 | 1127 | 708 | 0 | 185 |
| C2 | Cube | 433 | 1127 | 686.76 | 21.24 | 185 |
| C3 | | 433 | 1127 | 644.28 | 63.27 | 185 |
| B1 | | 433 | 1127 | 708 | 0 | 185 |
| B2 | Beam | 433 | 1127 | 686.76 | 21.24 | 185 |
| B3 | | 433 | 1127 | 644.28 | 63.27 | 185 |

As graphite powder can result in the poor workability of fresh concrete, polycarboxylic acid superplasticizer with a water reducing ratio of 30% was added to provide proper fluidity. Fine aggregates of river sand, with a fineness modulus of 2.8, and coarse aggregates of crushed stone, with a grain size of 5–25 mm, were used. The strength grade of C1 and B1 was C30. Previous studies have shown that while ensuring the mechanical and electrical properties of graphite concrete, the content of graphite should be less than 10% [13,14]. Therefore, in this study, the graphite powder content used was 3% and 9% respectively. Graphite powder was regarded as part of the fine aggregate.

*2.2. Test Procedure*

The cube-shaped and beam-shaped concrete specimens were cured at a temperature of about 20 ± 2 °C and >95% relative humidity for 28 days. The uniaxial compression tests of cube-shaped specimens, conducted in accordance with Chinese standard GB/T 50081-2019, were carried out on a WAW-1000 microcomputer-controlled electro-hydraulic servo universal testing machine, manufactured by Sansi Zongheng Machinery Manufacturing Co., Ltd., Shanghai, China, using a stress-controlled constant loading rate method with a loading rate of 0.05 MPa/s [15], as shown in Figure 3. For each cube-shaped group, three

cubic specimens were used and the mean value was reported as the compressive strength result. The three-point bending tests of beam-shaped specimens, conducted in accordance with DL/T 5332-2005, were performed under a constant loading rate of 10 N/s before the ultimate load of 40%, and then loaded to failure at 3 N/s. The crack mouth opening displacement (CMOD) corresponding to the applied load was measured by YYU-10/80 clamp extensometer, manufactured by NCS Testing Technology Co., Ltd., Beijing, China as shown in Figure 4. On the opposite side, four strain gauges were spaced at equal intervals on the specimen, as shown in Figure 5. There were also three specimens for each beam-shaped group. When the loading stress–CMOD curve was compared with ultimate load, initial fracture toughness, and unstable fracture toughness, if the differences between different test blocks in the same group were significant, the specimen with the middle performance value was used to represent this group.

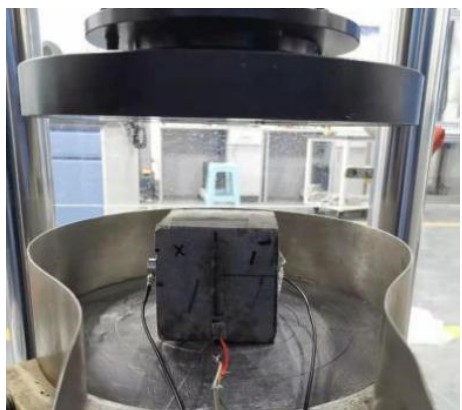

**Figure 3.** UCS test with AE.

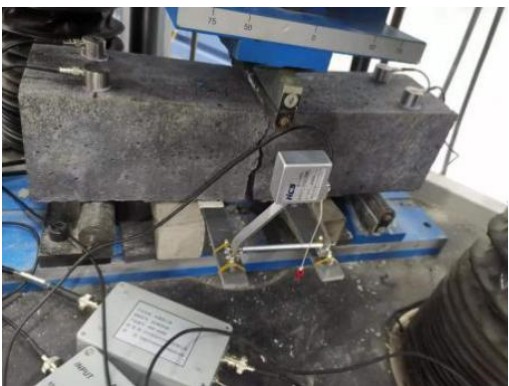

**Figure 4.** TPB test with AE.

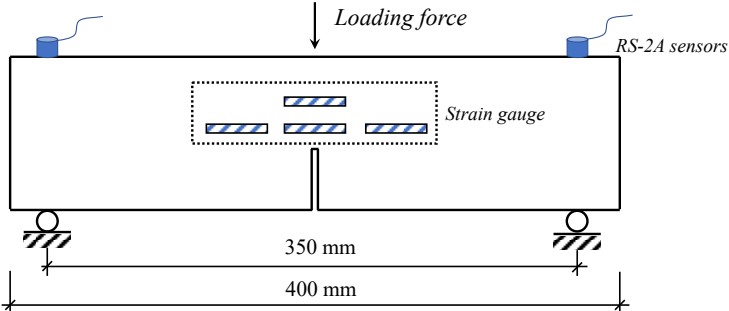

**Figure 5.** Schematic diagram of the three-point bending test of concrete beam with precast notch.

During the loading processes of uniaxial compressive test and three-point bending test, DHDAS dynamic data acquisition and analysis system were used to collect the strain; a DS5-8B acoustic emission detector with four RS-2A sensors (produced by Beijing Soft Island Times Technology Co., Ltd., Beijing, China) was used to record acoustic emission signals. The frequency range of RS-2A sensor is 50–400 kHz, and the resonance frequency is 150 kHz. The acoustic emission threshold of this experiment was set to 45 dB. The external preamplifier and main amplifier gain were both set at a fixed level of 40 dB.

Generally, there are two ways to obtain the initiation time of a crack during the three-point bending test: the F-CMOD curve turning point method [16] and the strain gauge turning point method [17]. Because the precision of strain gauge turning point method is better [17], the initial cracking load was determined by means of resistance strain gauge.

## 3. Results and Analysis

### 3.1. Mechanical Performance

#### 3.1.1. Compressive Strength Performance

The 28 d compressive strength of concrete containing varying amounts of graphite powder are shown in Figure 6. It can be seen that the compressive strength is inversely related to graphite powder content, decreased by 10.70% when the graphite powder content was raised from 0% to 3%, and decreased by 30.17% when raised to 9%. The compact structure of the hydration products is an essential guarantee for the formation of high-strength cement-based materials [18,19]. First, due to the poor hydrophilicity of graphite powder [20], it is difficult for the hydration products of cement to adhere on its surface, and the interfacial bonding between cement stone and graphite powder is weak [14]. Second, graphite powder is composed of series of stacked parallel layer planes, and within each layer plane, the carbon atom is bonded to three others, forming a series of continuous hexagons in what can be considered as an essentially infinite two-dimensional molecule [21], so its surface is extraordinary flat, resulting in a low interlocking force between cement stone and graphite powder. Owing to the poor interfacial bonding, low interlocking force and the flake shape, the distribution of graphite powder causes lots of initial defects to concrete, and the strength of the specimen is reduced accordingly. To reduce or eradicate the negative effects of graphite powder on compressive strength, polyvinyl alcohol (PVA) fibers are one of the most economical and effective ways [22].

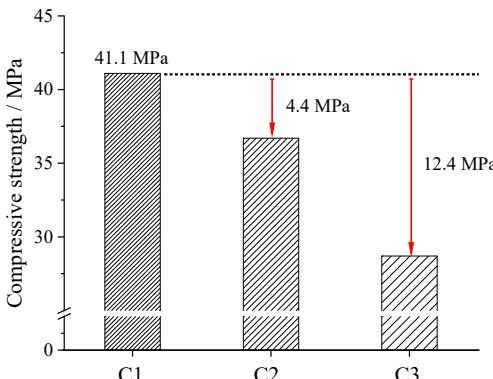

**Figure 6.** Compressive strength of concrete with different amount of graphite powder.

The stress–strain curves of concrete with different content of graphite powder are shown in Figure 7. The red arrow represents the reduction of compressive strength compared with group C1. As shown, due to the insufficient stiffness of the loading machine, the descending sections of the specimens are not obvious. When the loading stress reaches the compressive strength of the specimen, the elastic strain energy gathered in the loading machine is larger than the strain energy absorbed by the concrete, leading to a sudden destruction of the specimen [23]. The stress–strain curve of the reference group C1, can be divided into the elastic and plastic stages. However, as the graphite powder content

increases, the elastic characteristics become less clear, implying that graphite powder may cause a lot of plastic damage to concrete under low stress. During the loading process, poor adhesion of the graphite and cement stone interface failed, lots of flake-shaped micro cracks occurred, resulting in relative slip occurring between graphite powder and cement stone. As a result, as the graphite powder content increases from 0% to 3% and 9%, the elastic module decreases from 28.8 to 21.5 and 15.9 GPa.



**Figure 7.** Stress–strain curve with different amounts of graphite powder.

In addition, the initial defects caused by graphite powder also change the failure pattern. As for C1, due to the cyclo-hoop effect, the surrounding concrete was crushed and spalled after maximum stress, and the specimen was pyramidal, as shown in Figure 8. However, the decrease in concrete elastic modulus and the increase in concrete initial damage caused by graphite powder weakens the lateral restraint of the indenters to inner concrete [24], resulting in a weakened cyclo-hoop effect, leading to sheet-shaped spalling failure, as shown in Figure 9.

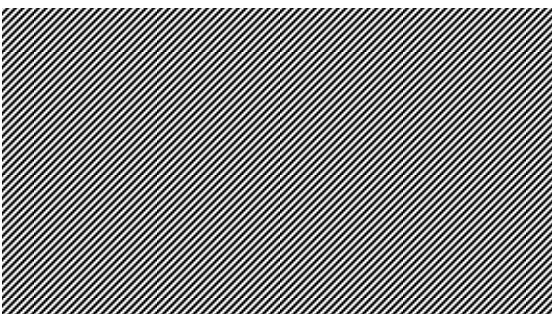

**Figure 8.** Failure pattern of C1. (**a**) appearance; (**b**) crack propagation.

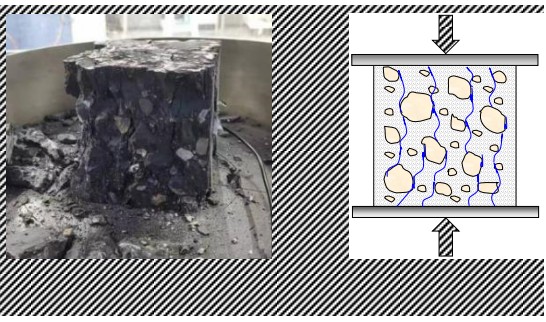

**Figure 9.** Failure pattern of C3. (**a**) appearance; (**b**) crack propagation.

3.1.2. Fracture Performance

As shown in Figure 10, although the shape of the loading stress-crack mouth opening displacement (F-CMOD) curves of the specimens with a different content of graphite powder makes a difference, the trends are similar. Each F-CMOD curve can be divided into

three stages, which are the linear rising stage, the curve rising stage, and the curve falling stage, usually representing the three periods of crack propagation: initial crack initiation, stable crack propagation, and instability failure [25]. It can be seen that the initial cracking load is more and more inversely related to graphite powder content, and was decreased by 8.7% when its content was increased from to 0% to 3%, and decreased by 34.8% when raised to 9%. Furthermore, compared to the reference group B1, group B2 and B3 exhibit a decrease of 4.9% and 8.7% of the maximum load, but their critical crack mouth opening displacement Vc are increased by 45.0% and 141.0%. This is largely due to the high stiffness of graphite powder, so even its adhesion with cement paste is weak; graphite powder can still play a significant role in limiting the expansion of cracks when it is not at a right angle to the cracks, as shown in Figure 11.

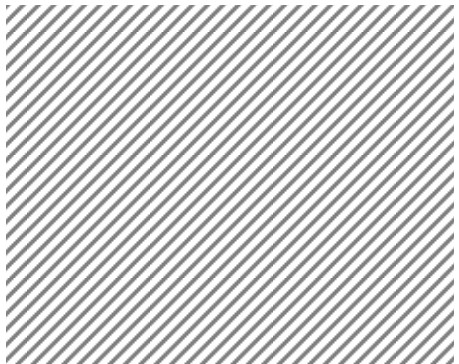

**Figure 10.** F-CMOD curves.

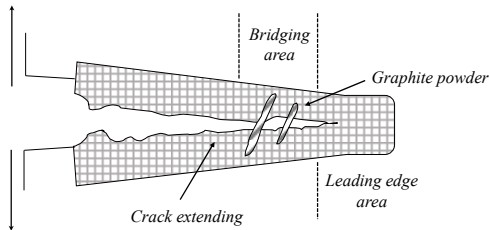

**Figure 11.** Schematic diagram of the bridging effect.

The double-K fracture criterion (including initial fracture toughness $K_{IC}^Q$ and unstable fracture toughness $K_{IC}^S$) proposed by Xu and Reinhardt [26,27] has been widely used to judge the fracture characteristics of concrete. The initial fracture toughness $K_{IC}^Q$ (MP·m$^{1/2}$), corresponding to the initial cracking load $F_Q$ (kN) and precast notch length $a_0$ (m), can be calculated by Equation (1). The unstable fracture toughness $K_{IC}^S$ (MP·m$^{1/2}$), corresponding to maximum load $F_{max}$ (kN) and critical crack $a_c$ (m), can be calculated by Equation (2).

$$K_{IC}^Q = \frac{1.5(F_Q + \frac{mg}{2} \times 10^{-2}) \times 10^{-3}S\sqrt{a_0}}{th^2}f(\alpha) \tag{1}$$

$$K_{IC}^S = \frac{1.5(F_{\max} + \frac{mg}{2} \times 10^{-2}) \times 10^{-3}S\sqrt{\alpha_c}}{th^2}f(\beta) \tag{2}$$

$$f(\alpha) = \frac{1.99 - \beta(1-\beta)(2.15 - 3.93\beta + 2.7\beta^2)}{(1+2\beta)(1-\beta)^{3/2}}, \alpha = \frac{a_0}{h} \tag{3}$$

$$f(\beta) = \frac{1.99 - a(1-a)(2.15 - 3.93a + 2.7a^2)}{(1+2a)(1-a)^{3/2}}, \beta = \frac{a_c}{h} \tag{4}$$

$$a_c = \frac{2}{\pi}(h + h_0)\arctan\sqrt{\frac{tEV_c}{32.6F_{\max}} - 0.1135} - h_0 \tag{5}$$

$$E = \frac{1}{tc_i}\left[3.70 + 32.6\tan^2\left(\frac{\pi}{2}\frac{a_0 + h_0}{h + h_0}\right)\right] \tag{6}$$

where $t$, $h$, and $L$(m) are the width, height, and span, respectively, of the precast notch beam; $m$ (kg) and $S$ (m) are the weight and span between the two supports; g (9.81 m/s$^2$) is the gravitational acceleration; $V_c$ (µm) is the critical crack mouth opening displacement; $h_0$ (m) is the thickness of the clip gauge holder; $c_i$ (µm/kN) is the initial compliance of the F-CMOD curve; and $E$ (GPa) is the elastic modulus predicted from the F-CMOD curve calculated by Equation (6).

Due to the non-standard dimension of the specimens, based on Weibull's statistical theory of brittle failure, the fracture toughness of standard dimension specimens can be obtained from Equation (7).

$$K_{IC}^{standard} = \left(\frac{V_{non-standard}}{V_{standard}}\right)^{1/\alpha}\left(\frac{h_{standard}}{h_{non-standard}}\right)^{1/2}K_{IC}^{non-standard} \tag{7}$$

where $\alpha$ is the Weibull coefficient, usually proposed between 7 and 13 (we used 10); $V_{standard}$ and $h_{standard}$ are the volume and height of a standard specimen, 0.024 m$^3$ and 0.2 m, respectively. In this study, the specimen volume $V_{non-standard}$ and the specimen height $h_{non-standard}$ are 0.004 m$^3$ and 0.1 m, respectively.

The calculation results of the initial fracture toughness, unstable fracture toughness, and critical crack of the specimens with different content of graphite powder are shown in Figure 12. It can be seen that the initial fracture toughness was reduced by 6.38% and 23.53%, basically in line with trend in the initial cracking load. On the contrary, under the influence of graphite powder, although the positive effect of initial cracking displacement on critical effective crack length was weakened by the significant decrease in elasticity modulus, and the maximum load was also decreased by 4.9% and 8.7%; an increase in the graphite powder content up to 3% resulted in an increase in the unstable fracture toughness of concrete up to 1.60 MPa·m$^{1/2}$ (by 1.7%); a further increase in the graphite powder content up to 9% resulted in an increase in unstable fracture toughness to 1.73 MPa·m$^{1/2}$, of 9.9%. This indicates that as a result of graphite powder partly replacing fine aggregate, the decreasing of initial fracture toughness and increasing of unstable fracture toughness improve the safety redundancy of concrete to some extent.

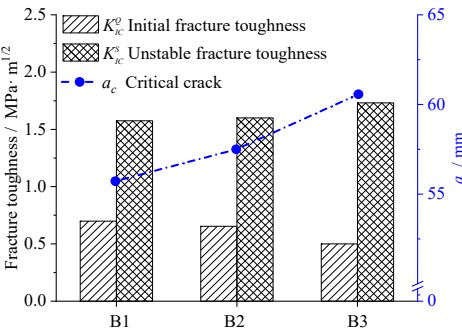

**Figure 12.** Fracture toughness and critical crack length.

*3.2. Acoustic Emission Characteristic*

3.2.1. Acoustic Emission Counts and Energy

(1)　Uniaxial compressive strength test

The relationships of cumulative ringing counts, cumulative energy release, and loading stress with time during the loading process of uniaxial compressive strength are present in Figure 13. The red chain-dotted lines are used to identify the first and the second turning points of AE signals, named as *M* and *N*, respectively.

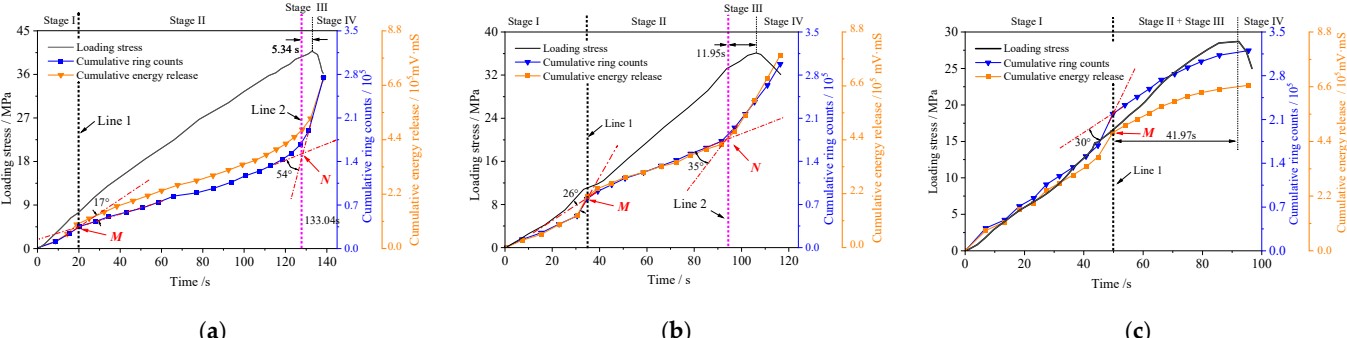

**Figure 13.** AE signal–time and stress–time curves of concrete with different graphite powder content during compressive tests. (**a**) C1; (**b**) C2; (**c**) C3.

As seen in Figure 13, cumulative ringing counts curve and cumulative energy release curve variation tendencies are approximate and unanimous. Based on the cumulative ringing counts curve, the process of C1 comprises four stages: initial compaction stage (Stage I), stable crack growth stage (Stage II), pre-peak stage (Stage III), and post-peak stage (Stage IV) [28]. For C1 shown in Figure 13a, in Stage I (~18% of the peak load), the cumulative ringing counts curve is not completely linear, but exhibits a little concave downward form. During the early loading of Stage I, the increasing-growing cumulative ringing counts and cumulative energy are from the microcrack closure of the concrete, and afterward, when the micro-cracks are compacted, and the increasing-growing trend is restrained. AE signals then go a little gently before proceeding to the next stage. At Stage II (18~95% of the peak load), the cumulative ringing counts and cumulative energy increase linearly with time, indicating the crack propagation was relative stable. After that, at the pre-peak region of Stage III (~95% of the peak load), with rapid progress in cumulative ringing counts and cumulative energy, there was large propagation of microcracks in the concrete, but almost no appreciable macroscopic crack on the surface. At the post-peak region of Stage IV, with the stress decreasing with time, the cumulative ringing counts and cumulative energy remain at a high level until rapid macroscopic crack propagation occurs. Thus, for concrete specimen without graphite powder, the turning point between Stage II and III (marked as Line #2) can be seen as a safety warning for concrete failure.

As shown in Figure 13b, when the graphite powder content was raised from 0% to 3%, the turning angle of cumulative ringing counts curve near Line #2 decreases from 54° to 35° and the failure time reduces from 132.83 s to 106.15 s, but the prewarning time formed by Line #2 increases significantly, from 5.34 s to 11.95 s. However, as shown in Figure 13c, when graphite powder content is further increased to 9%, the turning angle nearby is hard to be found, because there is no obvious boundary between Stage II and III. This suggests that, under the action of uniaxial pressure, a low dosage of graphite powder can extend the prewarning time of Line #2, but an excessively high content will reduce its significant degree, making it no longer valid.

By comparing Figure 13a–c, it also can be found that with increasing graphite powder content delaying the boundary between Stage I and Stage II (marked as Line #1)—up from 19.87 s in C1 to 34.95 s in C2, and after, 49.90 s in C3. The stress at Line #1 is increased from 6.23 MPa to 11.25 MPa, and then to 16.61 MPa, indicating that the microcracks caused by graphite powder require more time and pressure to close. Furthermore, as the graphite powder content increases, so does the turning angle of cumulative ringing counts curve near Line #1, increasing from 17° to 26° and 30°, making it easier to identify signal changes around Line #1.

In summary, when the graphite powder content is low, there are two sharp raises on the curve of cumulative ringing counts or cumulative energy release curve, and the second sharp raise can be seen as the prewarning signal for concrete failure, whereas when the graphite powder content is high and there is only one turning point on the curve, the only sharp raise can be seen as the prewarning signal for concrete failure. Give that 3% graphite

powder not only provides a reliable safety warning for compressive failure, but also has an acceptable negative influence on strength, when used in conjunction with other conductive filler, such dosage is suggested.

(2)   Three-point bending test

The relationships of AE ringing count, energy release, and loading stress with time during the loading process of three-point bending test are presented in Figure 14. The first and the second turning points of AE signals are marked as *K* and *L*, respectively.

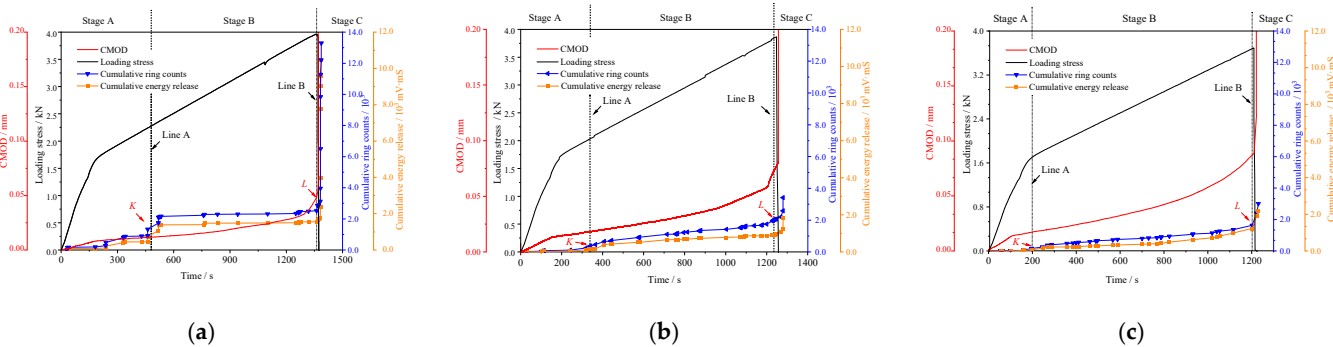

**Figure 14.** AE signal–time and stress–time curves of concrete with different graphite powder content during three-point bending test. (**a**) B1; (**b**) B2; (**c**) B3.

As shown in Figure 14a, during three-point bending test, there are also two obvious turning points on the AE curves; one representing initial cracking (marked as *K*), the other representing unstable cracking (marked as *L*); hereby, the failure process during three-point bending test can be roughly divided into three stages: the initial silent stage (Stage A), the intermediate stable stage (Stage B), and the last peak stage (Stage C). While cumulative ringing counts curve shows a similar trend as that for cumulative energy; it changes more rapidly near turning points, suggesting that cumulative ringing counts curve is more suitable for describing the failure process during three-point bending tests [29].

In the beginning of the initial silent period, only a few of AE signals are observed, indicating that almost no cracks are generated, and the internal damage is extremely minor. At the end of the first stage, the damage inside concrete is further increased with the increase in of loading stress, and the first sharp rise of cumulative ringing counts appears, suggesting a small number of discrete microcracks develops at the tip of the precast notch. After entering the second stage (stable period), as a consequence of the stable expansion of cracks, the activity of AE signals is reduced until another sharp rise in damage, where both the ringing counts and the strain energy released by the fracture rise sharply. The second turning point appears slightly earlier than the peak load, and the identification point of failure precursor information can provide a safety warning for the development of concrete fracture. During the last stage (peak period), the AE ringing counts and energy released increase rapidly and reach an obvious peak value, and the crack enters a rapid expansion period until failure.

As discussed before, in the tension zone, the initial defects caused by graphite powder are partially offset by its bridging effect, thus the maximum load is just slightly reduced by graphite powder. Due to the better performance of the graphite powder bridging effect at large deformations, with graphite powder content increasing, although the initial cracking time is advanced, the crack mouth opening displacement (CMOD) is larger under the same load.

During the fracture process, the AE activity of concrete in the tensile zone is reduced by the initial defects caused by graphite powder. Moreover, the bridging effect of graphite powder is a result of its high stiffness, not its interfacial bonding with cement stone, so the pulling process of graphite powder out of cement stone generates less AE signals. As a result, the amount of AE activity during loading is significantly reduced by graphite

powder, which is consistent with the findings of Ref. [12]. The AE signals around turning point *L* and *K* in Figure 14b,c are obviously less active than Figure 14a. However, after adding graphite powder, because of the enhancement of AE activity in the compressive zone, the AE signals between *L* and *K* are significantly increased.

3.2.2. Damage Evolution

It is well known that acoustic emission signals are directly related to the damage process, and can be used to characterize the damage evolution of materials [30,31]. Since the developments of cracks in concrete are irregularly, based on continuum damage mechanics, the meso micro unit strength of concrete is assumed to follow a Weibull distribution [32], and its distribution density function can be calculated as follows:

$$\varphi(x) = \frac{n}{m}\left(\frac{x}{m}\right)^{n-1} \exp\left[-\left(\frac{x}{m}\right)^n\right] \tag{8}$$

where *x* is the random variable ($x \geq 0$), *m* and *n* are the scale parameter and shape parameter of the Weibull distribution.

According to damage mechanics theory, damage value D also represents the damage degree of the materials; its relationship between distribution density function is as follows [33]:

$$\frac{dD}{dx} = \varphi(x) \tag{9}$$

Thus,

$$D = \int_0^x \varphi(x)dx = 1 - \exp\left[-\left(\frac{x}{m}\right)^n\right] \tag{10}$$

As ringing counts are highly associated with deterioration and directly reflect the internal damage, their pattern will be inevitably consistent with the statistical distribution of internal defects in the materials. By setting the total counts as $R_T$ when the specimen was loaded to failure, the AE accumulative counts *R* can be expressed as follows:

$$R = R_T \int_0^x \varphi(x)dx \tag{11}$$

Combining Equation (8) with Equation (11), it can be found that:

$$\frac{R}{R_T} = 1 - \exp\left[-\left(\frac{x}{m}\right)^n\right] \tag{12}$$

After comparing Equations (10) and (12), it was found that the damage value *D* can be described as a function of ring counts:

$$D = \frac{R}{R_T} \tag{13}$$

Figure 15 plots the curves of damage value *D* with strain $\varepsilon$ for the specimens with different graphite powder content under the compression test. The red chain-dotted lines are used to identify the turning points of damage value, named as *O*.

It can be seen from Figure 15a, because of the low density of original defects, $D$-$\varepsilon$ relation of C1 is almost linear before failure. At the beginning of loading, the damage value *D* slowly increases with the increasing stress as the original cracks and pores compacts. The damage value suddenly accelerates after strain has reached 97.7% of the peak strain (marked as *O*). Afterwards, within only 2.3% of strain, the damage degree increased from 0.61 to 1.0, indicating typical brittle fracture behavior. As shown in Figure 15b,c, due to the initial defects caused by graphite powder [12], although a high damage value of concrete occurs at low strain levels, the strain between turning point *O* and peak load increases to 10.6% and 31.5%; the deformation is larger in failure, and the *D*–$\varepsilon$ curve has a more linear

relationship, so that the damage form is changed from a brittle fracture to ductile fracture, and the failure of concrete under compression is more predictable.

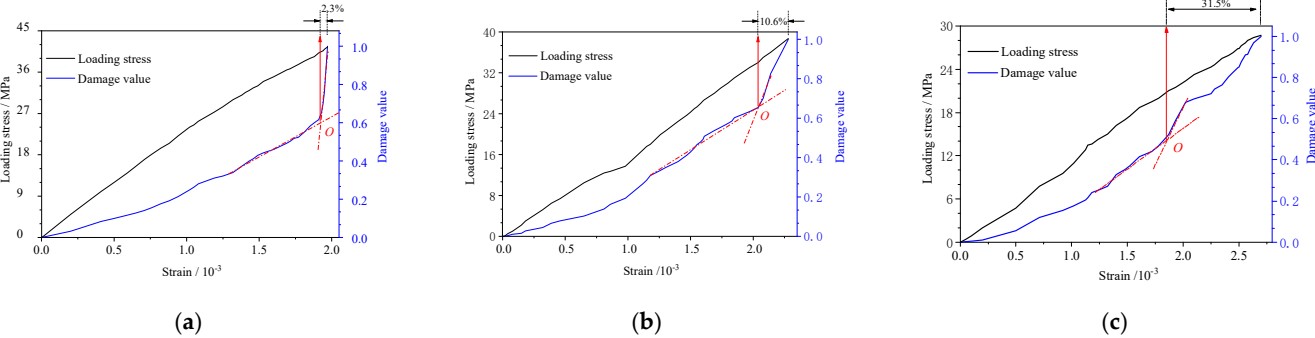

**Figure 15.** The curves of AE damage variable with strain under UCS test. (**a**) C1; (**b**) C2; (**c**) C3.

Figure 16 plots the curves of damage value *D* with crack mouth opening displacement *CMOD* for the specimens with different content of graphite powder under three-point bending test. The red chain-dotted lines are used to identify the turning points of damage value, named as *P*.

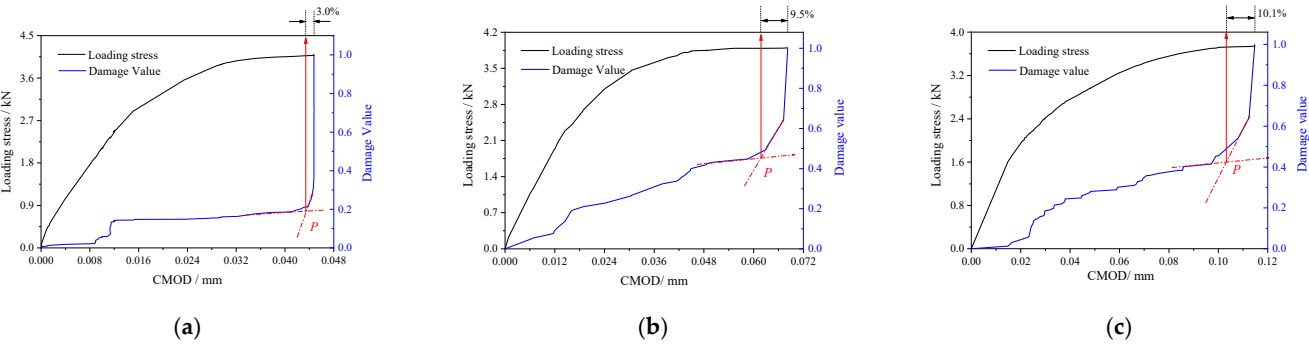

**Figure 16.** The curve of damage variable with load under TPB tests. (**a**) B1; (**b**) B2; (**c**) B3.

During the three-point bending test, the failure process of concrete without graphite powder exhibits obvious step characteristics, as shown in Figure 16a. After a sharp increase in damage between 7.91 μm and 15.39 μm, there was a long period of silence, with almost no damage produced until failure. However, as shown in Figure 16b,c, after the addition of graphite powder, damage value continues to grow during the whole fracture process, making the process of damage development easier to detect.

## 4. Conclusions

(1) Graphite powder causes obvious initial defects to concrete, which not only leads to a decrease in concrete integrity, but also reduces the elasticity modulus and the restraining action of the indenter on the internal concrete, further resulting in a decrease in compressive strength, reduced by 10.7% at a graphite powder dosage of 3%, and by 30.17% at a graphite powder of 9%. In addition, with increasing graphite powder, as a result of the increased initial defects to concrete and the decreasing cyclo-hoop effect of the indenter on internal concrete, the failure pattern of specimens in uniaxial compressive strength tests changed from block failure to a sheet-shaped spalling one, making them easier to observe.

(2) During the compressive process, there are two sharp raises on the cumulative ringing counts curve, or cumulative energy curve, of the specimen without graphite powder. As the graphite powder content increases, the time of the second sharp raise advances and its significance gradually decreases, whereas the time of the first sharp raise is

delayed and its significance increases. As a result, if the graphite powder content is low, the second sharp raise is more suitable for predicting failure, whereas if the content is high, the first sharp raise is more suitable. Regarding the second sharp raise as the failure alarm, 3% graphite powder can increase the warning time from 4.01% ahead of failure to 12.24%. If the graphite powder content reaches 9% and the first sharp raise is regarded as the failure alarm, the warning time will be further increased to 45.71%.

(3) There are two main effects of graphite powder on concrete in the fracture process: damaging and bridging. Because the bridging effect is stronger at large deformations, graphite powder causes a significant reduction in initial cracking load, but not in ultimate load. Furthermore, as the graphite powder content increases, the elasticity modulus and ultimate load stress decreases, but the critical crack mouth opening displacement grew significantly due to the bridging effect, leading to a slight rise in unstable fracture toughness.

(4) As for specimens without graphite powder, there are also two sharp raises on AE curves during the fracture process, similar to UCS tests. However, its AE signals cannot be used as a safety alarm, because there is a long period of silence before the second sharp raise, and the second sharp raise is too close to the ultimate load. In contrast, AE signals from the concrete are continuously generated before failure when mixed with graphite powder, allowing the development of the failure process to be identified.

(5) The damage value calculated by acoustic emission ringing counts shows that graphite causes damage to concrete at low loads, both in compressive and flexural processes. Therefore, the damage process of graphite concrete can be suitably described by acoustic emission.

**Author Contributions:** Project administration, W.H. (Wei He); Supervision, W.H. (Wei He); Writing—original draft, W.H. (Wei He) and W.H. (Wenru Hao); Writing—review & editing, W.H. (Wei He) and X.M.; Data curation, W.H. (Wenru Hao) and X.S.; Methodology, X.M. and Y.S.; Investigation, P.Z. and W.H. (Wenru Hao); Resources, Y.S. and X.M.; Funding acquisition, W.H. (Wei He) and P.Z. All authors have read and agreed to the published version of the manuscript.

**Funding:** This research was funded by Funded by Beijing Municipal Natural Science Foundation (NO.8204058), National Natural Science Foundation of China (Grant NO.51908022), China Postdoctoral Science Foundation (Grant NO.2019M660501), and Program for Scientific Research of Beijing Education Commission (Grant No.KM202110016013). The Fundamental Research Funds for Beijing University of Civil Engineering and Architecture (X20032).

**Institutional Review Board Statement:** Not applicable.

**Informed Consent Statement:** Not applicable.

**Data Availability Statement:** The data in this paper are given in the tables and figures within the manuscript.

**Conflicts of Interest:** The authors declare no conflict of interest.

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
