# Peer review of "Influence of Graphite Powder on the Mechanical and Acoustic Emission Characteristics of Concrete"

_buildings, doi:10.3390/buildings12010018_

Round 1

Reviewer 1 Report

This paper examines the influence of graphite powder on the mechanical and AE characteristics of concrete from compression test and three-point bending test. The investigations are interesting to readers of this journal. The following suggestions are needed to be addressed:

(1) To improve the quality of this manuscript, authors should properly edit the manuscript. Authors should pay attention to English grammar, spelling and sentence structure so that the goals and results of the present method can be clear to the reader. As follows:

[1] the word “showed” in line 7 of page 1 should be “show”;

[2] the word “By” in line 37 of page 1 loses the punctuation;

[3] the word “are” in line 46 of page 2 should be deleted;

[4] the word “Acoustic” in line 46 of page 2 should be changed;

[5] the word “used” in line 47 of page 2 should be “are used”;

[6] the word “B2 and B3 group” in line 173 of page 6 should be “groups B2 and B3”; [7] the word “Basing” in line 234 of page 8 should be “Based”;

[8] the word “present” in line 277 of page 9 should be “presented”;

[9] the word “Acoustic” in line 319 of page 10 should be revised;

[10] the words “Combing with” in line 338 of page 11 should be changed as “Combing”;

[11] Table 2: “Density(cm3/g)” should be replaced with “Density(g/cm3)”.

(2) The major problem of this manuscript is that many of the references are old and in Chinese. The Buildings in an international famous journal whose potential readers are not Chinese, so it is improper to cite many Chinese references, such as 14, 18, 19, 20, 28, 29 and so on. Besides, very few references are published after 2016. The authors should cite new findings to tell the readers what is new in this field. I think the following related ones can improve, such as Fractal analysis on pore structure and hydration of magnesium oxysulfate cements by first principle, thermodynamic and microstructure-based methods; Investigation and application of fractal theory in cement-based materials: A review.

(3) Page 9, Line 286: To maintain consistency, the second and third stages of the three-point bending test should be referred to as Stage B and C, same as shown in Fig.14.

(4) Page 11, Line 338: Equation (12) is derived by merging Equations (8) and (11).

(5) In the section of References, the forms of Refs. [2], [7], [20], [25], [27] and [28] are not the same as other articles. Authors should check carefully this section to maintain the consistency.

Author Response

Response to Reviewer 1 Comments

Point 1: To improve the quality of this manuscript, authors should properly edit the manuscript. Authors should pay attention to English grammar, spelling and sentence structure so that the goals and results of the present method can be clear to the reader. As follows:

[1] the word “showed” in line 7 of page 1 should be “show”;

[2] the word “By” in line 37 of page 1 loses the punctuation;

[3] the word “are” in line 46 of page 2 should be deleted;

[4] the word “Acoustic” in line 46 of page 2 should be changed;

[5] the word “used” in line 47 of page 2 should be “are used”;

[6] the word “B2 and B3 group” in line 173 of page 6 should be “groups B2 and B3”; [7] the word “Basing” in line 234 of page 8 should be “Based”;

[8] the word “present” in line 277 of page 9 should be “presented”;

[9] the word “Acoustic” in line 319 of page 10 should be revised;

[10] the words “Combing with” in line 338 of page 11 should be changed as “Combing”;

[11] Table 2: “Density(cm3/g)” should be replaced with “Density(g/cm3)”.

Response 1: Thank you for your suggestions. Those sentences have been revised as the reviewer suggested.

Point 2: The major problem of this manuscript is that many of the references are old and in Chinese. The Buildings in an international famous journal whose potential readers are not Chinese, so it is improper to cite many Chinese references, such as 14, 18, 19, 20, 28, 29 and so on. Besides, very few references are published after 2016. The authors should cite new findings to tell the readers what is new in this field. I think the following related ones can improve, such as Fractal analysis on pore structure and hydration of magnesium oxysulfate cements by first principle, thermodynamic and microstructure-based methods; Investigation and application of fractal theory in cement-based materials: A review.

Response 2: Thank you for the constructive suggestions. In the updated version, only Refs [11] and [16] are cited from Chinese magazines, while Refs [2], [4], [7], [8], [9], [12], [15], [18], [19], [24], [25], [28], [29], [30], [33], [34], most of which were published during the last three years, are used to replace older ones.

Point 3: Page 9, Line 286: To maintain consistency, the second and third stages of the three-point bending test should be referred to as Stage B and C, same as shown in Fig.14.

Response 3: Yes, Thank you. the sentence has been revised as the reviewer suggested.

Point 4: Page 11, Line 338: Equation (12) is derived by merging Equations (8) and (11).

Response 4: Thank you for pointing it out.  The sentence has been revised as the reviewer suggested.

Point 5: In the section of References, the forms of Refs. [2], [7], [20], [25], [27] and [28] are not the same as other articles. Authors should check carefully this section to maintain the consistency.

Response 5: Thank you for pointing them out. All of the Refs. have been carefully checked and revised.

Reviewer 2 Report

Very good paper on actual topic.

In the case of three-point bending test (Fig. 5), the side strain gauges will be more significantly affected by shear stresses. Has this effect been taken into account when processing?

A recommendation by the authors regarding the amount of graphite powder that seems acceptable should be added, e.i. where is the border when advantages still outweigh the disadvantages.

Why some authors are highlighted in yellow in references?

Between number and the physical units space shoud be placed (see for "seconds" unit in lines 255,256,264)

Author Response

Response to Reviewer 2 Comments

Point 1: Very good paper on actual topic. In the case of three-point bending test (Fig. 5), the side strain gauges will be more significantly affected by shear stresses. Has this effect been taken into account when processing?

Response 1: Thank you for your question. The two side strain gauges are indeed affected by shear stress to some extent, and its influence is almost linear with loading stress before initial cracking. However, for one thing, the shear stress is much lower than tensile stress. For another, the primary propose of the side strain gauges is to provide a hysteretic curve of Loading stress-gauge strain to determine initial cracking, and the linear increasing shear stress has little effect on the turning point of the hysteretic curve, so it can be ignored.

Point 2: A recommendation by the authors regarding the amount of graphite powder that seems acceptable should be added, e.i. where is the border when advantages still outweigh the disadvantages.

Response 2: Thank you for your construction suggestion. For compressive strength, a dosage of 3% graphite powder results in a 10.7% reduction, and there is around 12.24% time for warning before damage, thus an optimum dosage of 3% is recommended. This has been described in the revised manuscript. In this paper, graphite powder is meant to be utilized as a conductive filler for ECC. When graphite powder is used alone, its threshold dosage is generally less than 10%. In three-point bending test, even when the dosage is increased to 9%, the unstable toughness continues to increase, whereas the initial cracking toughness and peak load are only slightly reduced. As a result, when fracture behavior is taken into account, the dosage can be up to threshold dosage.

Point 3: Why some authors are highlighted in yellow in references?

Response 3: Thank you for point it out. It was a mistake, and has been corrected in the revise manuscript.

Point 4: Between number and the physical units space shoud be placed (see for "seconds" unit in lines 255,256,264)

Response 4: Thank you. It has been revised as the reviewer suggested.

Reviewer 3 Report

The article shows an interesting issue. It is undoubtedly a valuable research material. However, there are some errors in the article that require improvement and some doubts that should be clarified for the work to be fully understood:

  1. The literature review is insufficient. No information on other graphite powder studies in emission characteristics of concrete and such do exist.
  2. There are underlines in References and in Table 2, please correct it .
  3. Please include in the article (Experimental details section) which samples (B or C) correspond to which sample size.
  4. Lines 78-79.
    Please correct the measurement units 
  5. Please state according to which standards / instructions were the tests carried out?
  6. How many samples of each series were there? Are the results given in the article average measurements? What were the standard deviations or error bars?
  7.  In the article (results and analysis section) there are few references and comparisons to the results of other researchers. The research results should be compared with the results of other researchers, thanks to which the conclusions would be more complete and the validity and research results would gain credibility. 

Author Response

Response to Reviewer 3 Comments

Point 1: The literature review is insufficient. No information on other graphite powder studies in emission characteristics of concrete and such do exist.

Response 1: Thank you for your constructive suggestion. I also thought this is important. However, acoustic emission characteristics of electrically conductive concrete with graphite powder is rarely reported. After a difficult search, only one related article, which is published in 2019, was found. This article has been cited and discussed in detail, as shown in Line 60-65.

Point 2: There are underlines in References and in Table 2, please correct it.

Response 2: Thank you for pointing out, it has been revised.

Point 3: Please include in the article (Experimental details section) which samples (B or C) correspond to which sample size.

Response 3: Thank you, the sample size of group B and C have explained in Tab.3.

Point 4: Lines 78-79. Please correct the measurement units 

Response 4: Thank you for pointing out, it has been revised.

Point 5: Please state according to which standards / instructions were the tests carried out?

Response 5: Thank you for pointing out. In the revised manuscript, the standards have been explained in Line 110-113.

Point 6: How many samples of each series were there? Are the results given in the article average measurements? What were the standard deviations or error bars?

Response 6: There are three specimens for each beam shaped group. Considering that loading stress-CMOD curve is compared with ultimate load, initial fracture toughness and un-stable fracture toughness, if the differences between different test blocks in the same group are not significant, the specimen with the middle performance is used to represent this group, like other articles, such as Ref [12]. For each cube-shaped group, three cubic specimens were used and the mean value was reported as a compressive strength result.

Point 7: In the article (results and analysis section) there are few references and comparisons to the results of other researchers. The research results should be compared with the results of other researchers, thanks to which the conclusions would be more complete and the validity and research results would gain credibility. 

Response 7: Thank you for pointing out, it has been revised.